**Data Availability Statement:** All relevant data are within the paper and its S1 Fig and S1–S7 Tables.

**Funding:** The project was supported by EuroStars (project number E!8421). EuroStars did not have

# A new sensitive and fast assay for the detection of EGFR mutations in liquid biopsies

**Steffen Grann Jensen**[1][☯], **Samantha Epistolio**[2][☯], **Cesilie Lind Madsen**[3], **Majbritt Hauge Kyneb**[4], **Alice Riva**[2], **Alessia Paganotti**[5], **Jessica Barizzi**[2], **Rasmus Koefoed Petersen**[3], **Michael Børgesen**[3], **Francesca Molinari**[2], **Renzo Boldorini**[5,6], **Jan Lorenzen**[4], **Erik Sørensen**[7], **Ulf Bech Christensen**[3‡]*, **Estrid Høgdall**[1‡]*, **Milo Frattini**[2‡]*

1 Department of Pathology, Herlev—Gentofte University Hospital, Herlev, Denmark, 2 Institute of Pathology, Locarno, Switzerland, 3 PentaBase Aps, Odense, Denmark, 4 Life Science Division, Danish Technological Institute, Aarhus, Denmark, 5 Department of Pathology, 'Maggiore della Carità' Hospital, Novara, Italy, 6 Department of Health Sciences, Universitá degli Studi del Piemonte Orientale "A. Avogadro", Novara, Italy, 7 Department of Clinical Immunology, Rigshospitalet, University of Copenhagen, Copenhagen, Denmark

☯ These authors contributed equally to this work.
‡ UBC, EH and MF also contributed equally to this work.
* ubc@PentaBase.com (UBC); Estrid.Hoegdall@regionh.dk (EH); milo.frattini@ti.ch (MF)

## Abstract

### Background

A major perspective for the use of circulating tumor DNA (ctDNA) in the clinical setting of non-small cell lung cancer (NSCLC) is expected as predictive factor for resistance and response to EGFR TKI therapy and, especially, as a non-invasive alternative to tissue biopsy. However, ctDNA is both highly fragmented and mostly low concentrated in plasma and serum. On this basis, it is important to use a platform characterized by high sensitivity and linear performance in the low concentration range. This motivated us to evaluate the newly developed and commercially available SensiScreen® EGFR Liquid assay platform (PentaBase) with regard to sensitivity, linearity, repeatability and accuracy and finally to compare it to our already implemented methods. The validation was made in three independent European laboratories using two cohorts on a total of 68 unique liquid biopsies.

### Results

Using artificial samples containing 1600 copies of WT DNA spiked with 50% - 0.1% of mutant copies across a seven—log dilution scale, we assessed the sensitivity, linearity, repeatability and accuracy for the p.T790M, p.L858R and exon 19 deletion assays of the SensiScreen® EGFR Liquid assay platform. The lowest value detectable ranged from 0.5% to 0.1% with $R^2 \geq 0,97$ indicating good linearity. High PCR efficiency was shown for all three assays. In 102 single PCRs each containing theoretical one copy of the mutant at initiating, assays showed repeatable positivity in 75.5% - 80.4% of reactions. At low ctDNA levels, as in plasma, the SensiScreen® EGFR Liquid assay platform showed better sensitivity than the Therascreen® EGFR platform (Qiagen) and equal performance to the ctEGFR Mutation

any additional role in the study design, data collection and analysis, decision to publish, or preparation of the manuscript. PentaBase and ICP played the major role in study design, data collection, and preparation of the manuscript, as mentioned in the authors contribution. ICP is the scientific leader of the consortium. The role played by the different authors is reported in the "Authors contribution" section.

**Competing interests:** UBC, RKP, MB, and CLM are employees of PentaBase ApS. The SensiScreen Liquid EGFR assay is now part of a marketed product portfolio of PentaBase ApS. This commercial affiliation does not alter our adherence to PLOS ONE policies on sharing data and materials.

**Abbreviations:** (ctDNA), circulating tumor DNA; (NSCLC), non-small cell lung cancer; (OS), overall survival; (EGFR), epidermal growth factor receptor; (TKIs), tyrosine kinase inhibitors; (cfDNA), circulating free DNA; (NGS), next generation sequencing; (RT-PCR), real-time PCR; (ddPCR), droplet digital PCR; (BEAMing qPCR), beads emulsion amplification and magnetics quantitative PCR; (WT), wild type; (INAs®), intercalating nucleic acids; (PGM), personal genome machine; (AC), adenocarcinoma; (ml), millilitre; (EQA), external quality assessment.

Detection Kit (EntroGen) and the IOT® Oncomine cell-free nucleic acids assay (Thermo Fisher Scientific) with 100% concordance at the sequence level.

## Conclusion

For profiling clinical plasma samples, characterized by low ctDNA abundance, the SensiScreen® EGFR Liquid assay is able to identify down to 1 copy of mutant alleles and with its high sensitivity, linearity and accuracy it may be a competitive platform of choice.

## Introduction

Non-small cell lung cancer (NSCLC) is the major cause of tumor related deaths worldwide [1, 2]. A significant improvement in the overall survival (OS) has been obtained by the introduction of targeted therapies, especially those directed against the Epidermal Growth Factor Receptor (EGFR) [3–5]. NSCLC tumor development is highly heterogeneous with new tumor clones arising over time and during treatment, causing tumor progression and resistance to the existing therapy [6, 7]. To increase OS, a plethora of EGFR-tyrosine kinase inhibitors (TKIs) have been developed, starting from 1st generation TKIs to the most recent 3rd generation inhibitor, osimertinib [8–13]. Osimertinib was designed in order to target the main mechanism of resistance to standard EGFR TKIs (i.e.: the point mutation p.T790M). Subsequently, it has demonstrated good activity also against all the other EGFR mutations (exons 18–21) and has recently been approved as first line treatment of NSCLC [13–16]. Since EGFR TKI therapies must be administered only to patients harbouring the specific EGFR mutations, associated to documented response to these drugs, a precise and sensitive molecular characterization of hotspots within the DNA sequence of the EGFR tyrosine kinase domain is of primary relevance [17, 18]. A situation complicated, in vast majority of cases, by the availability of only low amounts of DNA for the molecular characterization of this type of cancer [19].

Due to localization, tumor resection is not possible in the majority of cases and lung biopsy sampling is challenging to obtain and painful to the patient [20, 21]. Moreover, a single tissue biopsy may not reflect the entire tumor heterogeneity, including the resistance clone of interest for planning of the correct treatment. In contrast, liquid biopsies, like blood samples, are believed better to reflect tumor heterogeneity and metastases, they are less invasive, and can be, quickly and easily accessed by a single needle stick. Thereby, liquid biopsies are fulfilling the huge requirement and focus upon translational less invasive monitoring of disease for qualification of personalized medicine [20–22]. Besides the introduction of TKIs, one of the most important improvements in the field of NSCLC has therefore been the introduction of liquid biopsy testing [23, 24].

A facing challenge by liquid biopsies is the detection of EGFR mutations at extremely low concentrations of circulating tumour DNA (ctDNA) (<1% in many cases) in a background noise of much higher, but overall low concentrations of circulating free DNA (cfDNA) originated from elsewhere than the tumour cells of interest [25]. Moreover, conditions in the blood are harsh for cfDNA with a reported half-life of only 10–16 minutes and ctDNA fragments are described as being shorter than non-malignant cfDNA [20, 26, 27]. This highlights the unambiguous need for highly sensitive and specific methodologies for ctDNA analysis [28–31]. Consequently, several assays have been developed to identify EGFR mutations using cfDNA as sample material for molecular profiling.

In general, targeted approaches require less cfDNA as input to obtain high analytic sensitivity than untargeted approaches (e.g.: next generation sequencing (NGS)) [20, 32]. This, despite strong efforts to improve detection limits [33]. The main targeted techniques are [30, 33]: Quantitative real-time PCR (RT-PCR) (e.g. Therascreen[®] (QIAGEN) [34], cobas[®] (Roche) [35, 36]), ctEGFR Mutation Detection Kit (EntroGen)) [37], droplet digital PCR (ddPCR) [19, 38, 39], beads emulsion amplification and magnetics quantitative PCR (BEAMing qPCR) [36] and ultra-deep NGS [40] (e.g. the Ion Torrent[®] based Oncomine™ Lung cfDNA Assay (Thermo Fisher Scientific)). All these approaches are characterized by different lowest values detectable [30, 41]. RT-PCR sensitivities varies from 5% down to 1% depending on the applied assay [36, 41–43]. The other aforementioned methodologies have the lowest values detectable that are equal to 0.1–1% or less. Due to sparse amounts of cfDNA extracted per plasma and serum sample, multiplex approaches promise to be more ideal and successful for ctDNA analysis than simplex based quantification platforms [25, 44, 45].

Altogether, this background motivated us to evaluate the performance of a new, highly sensitive and easy-to-use RT-PCR-based platform—the SensiScreen[®] CE IVD EGFR Liquid kit (Multiplex/Simplex) from PentaBase (Odense, Denmark)—promising identification of down to a single copy of EGFR mutant ctDNA in background of wild type (WT) DNA. Using WT DNA spiked with varying concentrations of mutated DNA from cell lines, we evaluated the sensitivity, linearity, specificity, repeatability and accuracy of the SensiScreen[®] EGFR Liquid assays. Next, performance was compared and evaluated to other RT-PCR based assays, the ctEGFR Mutation Detection Kit (EntroGen) and the Therascreen[®] EGFR RGQ PCR Kit (Qiagen) as well as the NGS based Oncomine™ Lung cfDNA Assay (Thermo Fisher Scientific) in three independent European laboratories.

## Materials and methods

### Cell line models

Genomic DNA from two EGFR mutant cell lines were used for evaluation of sensitivities and specificities of the SensiScreen[®] EGFR Liquid assays. Cell lines were subcultured in appropriate media with conditions according to the manufacturer's instructions (ATCC®—NCI-H1650 and NCI-H1975). Genomic DNA was isolated using the QIAamp Mini kit (Cat.no.51304) (Qiagen, Chatsworth, CA, USA) according to manufacturer's guidelines.

### Clinical patient samples

Liquid biopsies from a total of 68 patients, previously diagnosed with lung adenocarcinoma, were used to evaluate the SensiScreen[®] EGFR Liquid assay performance in a clinical setup (S1A and S1B Table). While cohort I was collected from 2016 to 2018, cohort II was collected between 2016 and 2019. For cohort I and II, the mean age at diagnosis was of 65.5 years (range: 37–84 years) and 69.5 years (range: 44–89 years), respectively (Fig 1). For cohort I, 15 patients were diagnosed with only the primary tumor, while the remaining 19 patients showed metastases in one or in multiple sites of the body (S1A Table). The study was approved by the Institutional Ethical Committee at the Institute of Pathology in Locarno, Switzerland and by the Central Denmark Region Committee on Health Research Ethics and by the Danish Data Protection Agency. All procedures were performed in accordance with the ethical standards of the Helsinki Declaration.

### The SensiScreen[®] EGFR Liquid kits

The RT-PCR-based SensiScreen[®] EGFR Liquid kits (PentaBase Aps, Odense, Denmark) (Cat. no. 5408/5409 / Cat.no. 3075/3076) are offered as a combination of simplex and multiplex

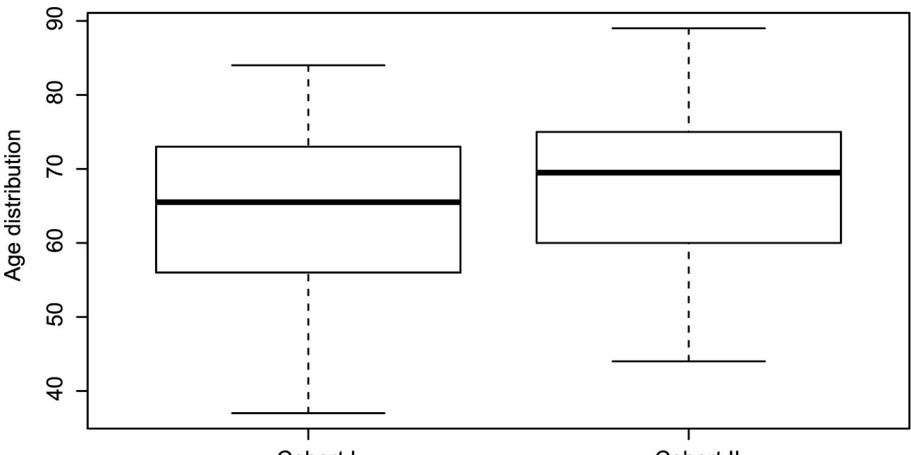

**Fig 1. Distribution of age for patients in cohort I (N = 34) and cohort II (N = 34).** Median age was 65.5 years for cohort I and 69.5 years for cohort II at the time of routine ctDNA EGFR analyses.

assays for the detection of mutations in EGFR exons 18, 19, 20 and 21 (S2 Table) using liquid biopsies as sample material. SensiScreen® EGFR Liquid assays were developed essentially as described for SensiScreen® assays for tumor tissue analysis by Christensen et al. [46] and Riva et al. [47]. The oligos used in the SensiScreen® EGFR Liquid kits are modified with intercalating nucleic acids (INAs®), also known as pentabases. Modified oligos include primers, probes and BaseBlockers™, where the BaseBlockers™ particularly block amplification of WT DNA as described previously [47]. The SensiScreen® EGFR Liquid kits contain a reference assay amplifying part of the EGFR gene by means of allele unspecific primers and a green fluorescent HydrolEasy™ probe. In addition, all reactions include an internal control assay containing allele-independent primers and a HEX-labelled HydrolEasy™ probe targeting the CYP17A1 gene, that does not interfere with amplification of the primary assays.

## SensiScreen® EGFR Liquid assay conditions

Evaluation of the SensiScreen® EGFR Liquid assay was performed in 25 μl total reaction volumes. The thermocycling conditions used were: 2 min of initial activation of the hotstart Taq-polymerase at 95˚C, followed by 45 cycles of a 2-step PCR with a 15 sec denaturation step at 94˚C and a 60 sec annealing and elongation step at 60˚C. Fluorescence was measured during or at the end of each elongation step. In order to make data analysis independent of the type of instrument used, the threshold was defined as 10% of the signal strength of the reference assay at cycle 45. Samples were considered valid when $29 < \text{Ct}_{\text{ref}} \leq 40$ and positive for mutation when $\text{Ct}_{\text{assay}} < 40$.

## Sensitivity, linearity, specificity, repeatability and accuracy studies

In order to evaluate the sensitivity and linearity of the SensiScreen® EGFR Liquid assays, serial dilutions of mutated cell line DNA in WT DNA (Cat.no. G3041) (Promega, Madison, WI, USA) were performed. SensiScreen® EGFR Liquid assays were tested using eight different concentrations of mutated template (50%, 10%, 5%, 2%, 1%, 0.5%, 0.1% and 0%) with a total DNA load of 5 ng. PCR analysis was performed for the exon 19 deletions, p.T790M and p. L858R according to the SensiScreen® protocol [48]. Dilutions were analysed on both reference and mutation specific assays in duplicates in two independent runs. Utilizing the MyGo Pro PCR software, the lowest value detectable and PCR efficiency were measured for the three

SensiScreen® EGFR Liquid assays. Ct values of the amplification plots were calculated by adjusting threshold to 10% of the amplification signal at cycle 45 for the reference assay and the lowest values detectable were derived from these. In vitro specificity of assays were analyzed using cell lines (H1975, carrying EGFR mutations T790M and L858R, and H1650, harbouring an EGFR exon 19 deletion) or linearized plasmid DNA containing EGFR mutations G719D, L861Q and L747S diluted 1:1 in WT cfDNA. Approximately 1600 copies of total DNA input were used for all experiments. To evaluate repeatable and accurate detection performance of the SensiScreen® EGFR Liquid assays at the low mutant copy level as are found in liquid biopsies, DNA from cell lines was diluted to concentrations of theoretically one mutant copy and approximately 1,600 copies of total WT DNA (Cat.no. G3041) (Promega) in 25 µl PCR reactions. Template dilutions were analysed in 102 replicates for each of the SensiScreen® EGFR Liquid assays (p.T790M, exon 19 deletions and p.L858R) according to the SensiScreen® protocol. For assessing the assay's background/noise we tested, in addition to WT genomic DNA, a pool of 58 age matched normal donors (i.e. samples assumed to not have the mutation present) with DNA isolated from plasma in order to evaluate the background (limit of blank, LoB) for each variant in the assay.

Analyses were performed utilizing MyGo Pro PCR Software v. 3.4.8 and R studio.

## Plasma and serum separation

For both cohorts, plasma and serum used in the study were surplus to requirements for previous diagnostic routine testing. All materials were anonymized.

Cohort I (plasma samples): From each patient, 2 x 9 ml of blood was collected in Cell-Free DNA BCT® tubes (Streck). After blood collection, the samples were turned upside down ten times and stored at room temperature until plasma and serum separation. Plasma separation was done within 48 hours after blood collection, by two centrifugation steps at 3000xg for 10 min followed by slow braking at room temperature. Before centrifugation the tubes were turned upside-down three times. Plasma was stored at -80˚C until DNA extraction or immediately used for DNA extraction.

Cohort I (serum samples): From each patient, 2 x 9 ml of blood was collected in Vacutainer® Plus Plastic Serum Tubes (Becton Dickinson, Franklin Lakes, New Jersey, USA), characterized by spray-coated silica with the purpose to activate coagulation. After blood collection, the samples were stored at room temperature until serum separation for at least one hour, in order to permit the clotting. Serum separation was done immediately after the coagulation time by one centrifugation step at 1000xg for 10 min at room temperature followed by slow braking. Serum was stored at -80˚C until DNA extraction or immediately used for DNA extraction.

Cohort II: From each patient, between 9 ml and 36 ml [Median;Mean/36;30.06] of blood were collected in 9 ml sodium citrate tubes. Samples were centrifuged at 2500xg for 10 min at 4˚C followed by slow braking, latest within one hour from blood collection. Plasma was collected immediately after centrifugation and either stored at -80˚C for a maximum of three days or immediately used for DNA extraction.

**Procedures for DNA extraction and mutational analysis by platforms (cohort I).** Cohort I analyses were performed at the University Hospital of Novara, Italy (A) and at the Institute of Pathology in Locarno, Switzerland (B).

*A. Therascreen® EGFR Plasma RGQ PCR Kit (Qiagen) (Cat.no. 874111)*: starting from 2 ml of plasma or serum, cfDNA was extracted using the QIAamp Circulating Nucleic Acid Kit (Qiagen) and stored at -20˚C. For each cfDNA sample, four tests were prepared: control reaction, p.T790M, p.L858R and exon 19 deletions. Real-time PCR was performed using 5 µl of

cfDNA according to manufacturer's guidelines in 0.1 ml tubes. Data analysis was performed using the Rotor-Gene Q Series Software 2.3 (Qiagen).

B. *IOT*® Oncomine *cell-free nucleic acids assays (Thermo Fisher Scientific, Waltham, MA USA) (Cat.no. A31149)*: starting from 4 ml of plasma or serum, cfDNA was extracted using the MagMax Cell-Free DNA Isolation Kit (Cat.no. A29319) (Thermo Fisher Scientific). DNA concentrations were determined using the Qubit dsDNA HS Assay Kit (Cat.no. Q32851) and the Qubit 2.0 instrument (Thermo Fisher Scientific). For library preparation, 10 ng of cfDNA was used. The libraries were quantified by qRT-PCR using the Ion Library TaqMan Quantitation Kit (Cat.no. 4468802) and 100pM were used for template preparation (Thermo Fisher Scientific). For template and sequencing, the Ion Personal Genome Machine (PGM) Hi-Q OT2 kit (Cat.no. A29900) and the Ion PGM system (Thermo Fisher Scientific) were used.

C. *SensiScreen*® *EGFR Liquid assays (PentaBase)*: 5 µl of cfDNA, either extracted using the QIAamp Circulating Nucleic Acid Kit (Qiagen) or the MagMax Cell-Free DNA Isolation Kit (Thermo Fisher Scientific), was added to the ready-to-use tubes for PCR reaction (Cat.no. 5408 and 5409). Real-Time PCR was performed using the CFX-96 instrument (Bio-Rad, Hercules, CA, USA) according to the SensiScreen® protocol.

**Procedures for DNA extraction and mutational analysis by platforms (cohort II).** Cohort II analyses were performed at the Department of Pathology, Herlev—Gentofte University Hospital, Denmark. cfDNA was extracted using the QIAamp Circulating Nucleic Acid Kit (Qiagen) according to the manufacturer's instructions. Samples with plasma volumes below 5 ml were diluted to 5 ml in PBS without $Ca^{++}$ and $Mg^{++}$ (Cat.no. 37350) (STEMCELL Technologies) prior to extraction. Elution buffer applied per column was mostly 30 µl but in few cases up to 50 µl. The total cfDNA concentrations were measured using Qubit dsDNA HS Assay Kit (Thermo Fisher Scientific).

*ctEGFR Mutation Detection Kit (Entrogen, Woodland Hills, CA, USA) (Cat.no. ctEGFR-48)*: either relative and/or absolute quantification of the EGFR mutations, c.2369C>T (p.T790M), exon 19 deletions, c.2573T>G (p.L858R) was performed. Samples were measured mostly in triplicate, some in duplicate, with no template control and high and/or low positive control provided with the kit included. For samples with low concentrations of cfDNA, only 1 µl of nuclease-free water was added to the PCR reaction.

*SensiScreen*® *EGFR Liquid assays (PentaBase)*: test of the SensiScreen® EGFR Liquid assays dispense-ready (Cat.no. 3075 and 3076), c.2369C>T (p.T790M), exon 19 deletions, c.2573T>G/c.2573_2574TG>GT (p.L858R), was done in single measurements due to sparse cfDNA amounts left after routine testing. Despite from running 50 PCR cycles instead of 45, RT-PCR conditions were performed according to the manufacturer's instructions on an ABI7500 (Thermo Fisher Scientific).

## Statistical analysis

Statistical analysis was carried out using Rstudio and Microsoft Office Excel. The expected distribution frequencies of samples diluted to an estimated one mutated copy per PCR reaction were calculated using the Poisson distribution probability function: $P(k) = e - \lambda(\lambda k/k!)$, where P is the probability of finding k mutations per PCR reaction and $\lambda$ is the expected average number of mutated copies per PCR reaction.

## Results

### Sensitivity and linearity

To enable assessment of sensitivity and linearity of the SensiScreen® EGFR Liquid assays, eight-point dilution series (1 ng/µl final DNA concentration) were made using WT DNA

spiked with different concentrations of DNA from mutant cell lines harbouring either the c.2235_2249del15 (p.E746_A750delELREA) exon 19 deletion, the c.2369C>T (p.T790M) mutation and/or the c.2573T>G (p.L858R) mutation (Fig 2).

All three assays were able to detect the investigated EGFR mutations at the highest dilution point of 0.1% when using standard analysis criteria. Notably, the 0.1% dilution point corresponds theoretically to only 1.6 (i.e. 1–2) copies of mutated DNA per PCR reaction based on 5 ng DNA input, assuming precise serial dilutions.

Next, we investigated the linearity of assays by assessing the coefficient of determination ($R^2$) for each assay across the dilution series (50% to 0.1%) (S3 Table). As expected, no detection was observed at no spike-in of mutated DNA in a background of WT DNA. Interestingly, all three assays showed high correlation, $R^2 \geq 0.97$ [49], and thereof good linearity across a 7-log scale ranging from 50% to 0.1% of mutated copies even though the variance observed for the 0.5% and 0.1% repetitions was significantly higher than the remaining dilution points (Fig 2 and S3 and S4 Tables). This is most likely due to the low number of mutated templates present in the PCR reactions. Furthermore, linear regression analysis of log-transformed mutated DNA inputs to PCR cycle thresholds revealed slopes for the best fitted line between -2.58 to -3.67 corresponding to PCR efficiencies of 87–144%, thereby suggesting acceptable efficiency of PCRs for the tested assays.

To ensure test specificity the assays ability to recognize similar but unrelated sequences were tested. Cell lines harbouring EGFR mutations or plasmids with the indicated mutated sequences diluted in WT cfDNA were analyzed, but none of the assays displayed valid signals except for the examined mutation (Fig 3 and S5 Table), indicating high specificity.

## Repeatability and accuracy at low mutant copy input level

The sensitivity studies indicated that the investigated SensiScreen® EGFR Liquid assays are able to detect a few or even just a single mutated copy of EGFR cell line template DNA in a background of WT DNA. To investigate this further, we assessed the capability of the three SensiScreen® EGFR Liquid target assays to repeatedly produce a mutation positive signal when using samples containing theoretically one copy of the specific mutated template in approx. 1,600 copies of WT DNA per PCR reaction in 102 replicates per assay (Fig 4). To assure that observed PCR signals were related to the mutation specific template, all conducted experiments included samples consisting of only WT DNA. Repeatability was defined as the ability of the assay to produce the same results for analyses of identical samples under the same conditions in the same laboratory. Using this setup, the c.2369C>T (p.T790M) assay gave a mutation-positive result in 80.4% (82/102) of the PCR reactions (Fig 4A and Table 1). For the exon 19 deletion assay and the c.2573T>G / c.2573_2574TG>GT (p.L858R) assay, the

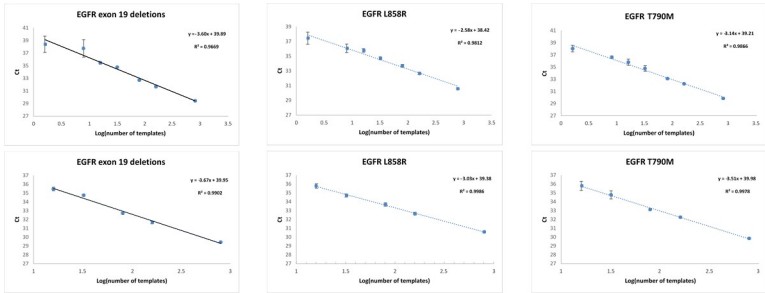

**Fig 2. Assessment of platform sensitivity, linearity and PCR efficiency for the SensiScreen® EGFR Liquid assays.** Logarithmic transformed DNA input (X-axis) according to detection cycle threshold (Y-axis) for assays.

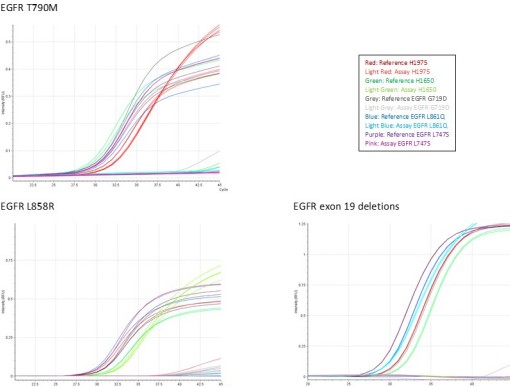

**Fig 3. Assessment of platform specificity.** PCR curves obtained when challenging assays with cell lines or plasmids containing the indicated EGFR mutations. Plasmids were diluted 1:1 in WT gDNA to ensure a total template amount of 1600 copies. Reference assay PCR curves are represented by heavy coloured lines and mutation specific assay PCR curves by light coloured lines as indicated in figure legends. Abbreviations: Ct, Cycle threshold; EGFR, Epidermal growth factor receptor.

**Fig 4. Test of repeatability at the low mutant copy level—estimation of the actual number of mutant copies in each of 102 aliquots containing theoretical 1 copy of spiked mutant.** A. Histograms showing the frequency and distribution of mutant copies detected in 102 replicates for assays. B. Boxplot showing the collectively recovered number of copies by the three assays and thereby the overall detected copy number by assays. Abbreviations: EGFR, Epidermal growth factor receptor.

**Table 1. Repeatability and accuracy evaluation of SensiScreen® EGFR Liquid assays: Number of copies mutated for Exon 19 deletion, T790M or L858R.**

| | Template | Number of mutated copies detected | | | | Total |
|---|---|---|---|---|---|---|
| | | 0 | 1 | 2 | ≥3 | |
| Exon 19 deletion | c.2235-2249del (Glu746-Ala750del) | 25 | 77 | | | 102 |
| T790M | c.2369C>T (p.T790M) | 20 | 82 | | | 102 |
| L858R | c.2573T>G (p.L858R) | 23 | 79 | | | 102 |
| Poisson (λ = 1) | NA | 37.5 | 64.5 | | | 102 |
| | | | 37.5 | 18.8 | 8.2 | |
| Poisson (λ = 1.5) | NA | 22.8 | 79.2 | | | 102 |
| | | | 34.1 | 25.6 | 19.5 | |

Abbreviations: EGFR, Epidermal growth factor receptor; NA, not available.

mutation-positive frequency was 75.5% (77/102) and 77.5% (79/102), respectively. All three SensiScreen® EGFR Liquid assays showed Gaussian distributions with means around 0.8 mutated copies indicating high accuracy but with small differences in precision among replicates (Fig 4A). Since the templates used were diluted to contain theoretically only a single mutated copy of the EGFR template, 36.8% of the PCR reactions would be expected to contain 0 mutated EGFR templates while 63,2% of the PCR reactions would be expected to contain 1 or more mutated EGFR templates when assuming that the distribution of mutated copies follows the Poisson model with a distribution rate parameter of 1 (Table 1). The fact that the fraction of samples with detected EGFR mutations is higher than expected, as compared to what will be expected when Poisson distributed, indicate that the concentration of mutated EGFR templates was slightly above 1 copy per PCR reaction. Hence, the data is more in line with a Poisson distribution rate parameter of 1.5. This does also explain the few outliers with >3 mutant copies (Fig 4A and 4B). In addition repeatability has been statistically confirmed through the analyses of fractile limits; that has been chosen according to theoretical number of zero samples and samples with more than 1 copy assuming a concentration of 1 copy/5 microL and Gaussian distribution (S6 Table). Thus, these data suggest that the three SensiScreen® EGFR Liquid assays are able to repeatedly detect a single mutated copy of EGFR template since 34 of the mutation-positive cases would still contain a single copy of mutated EGFR template when using a distribution rate parameter of 1.5 (Table 1).

The assessment of the LoB for each variant of the SensiScreen® EGFR Liquid assays (p. T790M, exon 19 deletions and p.L858R), performed by application of 20 replica of a pool of cfDNA from 58 healthy donors, revealed that none of these samples gave rise to valid PCR signals with Ct values ≤ 40 (Fig 5A–5C).

## Test of the SensiScreen® EGFR Liquid assays in clinical samples

To validate the SensiScreen® EGFR Liquid assays - c.2369C>T (p.T790M), exon 19 deletions, c.2573T>G/c.2573_2574TG>GT (p.L858R)—in the clinical setting, we used cfDNA extracted from blood of two retrospective cohorts including lung adenocarcinoma (AC) patients with or without metastases (S1A and S1B Table). Except for three serum samples included in cohort I, all cfDNA was extracted from plasma.

As generally seen for liquid biopsy samples [50–52], the amount of total cfDNA extracted per millilitre (ml) of plasma was low (Fig 6). For cohort I, the median amount of DNA extracted per ml of plasma was 11.30 ng [2.33ng/ml;86.28ng/ml]. For cohort II, the median level of DNA per ml of plasma was 15.19 ng [3.75ng/ml;81.67ng/ml]. DNA extracted from the three serum samples in cohort I was 5.07 ng/ml, 7.75 ng/ml and 34.11 ng/ml, respectively. The

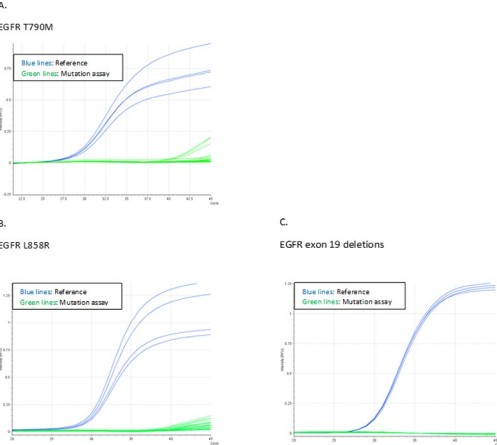

**Fig 5. Test of LoB.** Graphs describing the LoB evaluation based on the characterization of repeated testing of cfDNA pooled from 58 healthy donors A. Curves representing the T790M LoB results. B. Curves representing the L858R LoB results. C. Curves representing the EGFR exon 19 deletion LoB results.

DNA extracted from the serum samples was characterized by concentrations included in the quantitative range of the DNA obtained from plasma, suggesting the absence of gross contamination from leukocytes in serum. For some of the samples previously analysed in the routine setting, either no DNA or less than used for the routine analysis was available for test of the SensiScreen® EGFR Liquid assays, because we used the material left from the diagnostic routine. The results obtained by the SensiScreen® EGFR Liquid assays were matched and compared with those previously obtained by alternative methods (S7A and S7B Table).

For both cohorts, equal agreement of 94.1% (32/34) between the SensiScreen® EGFR Liquid assays to the Therascreen® (QIAGEN) and to the ctEGFR Mutation Detection Kit (EntroGen), respectively was observed (Tables 2 and 3). For cohort I, the SensiScreen® EGFR Liquid

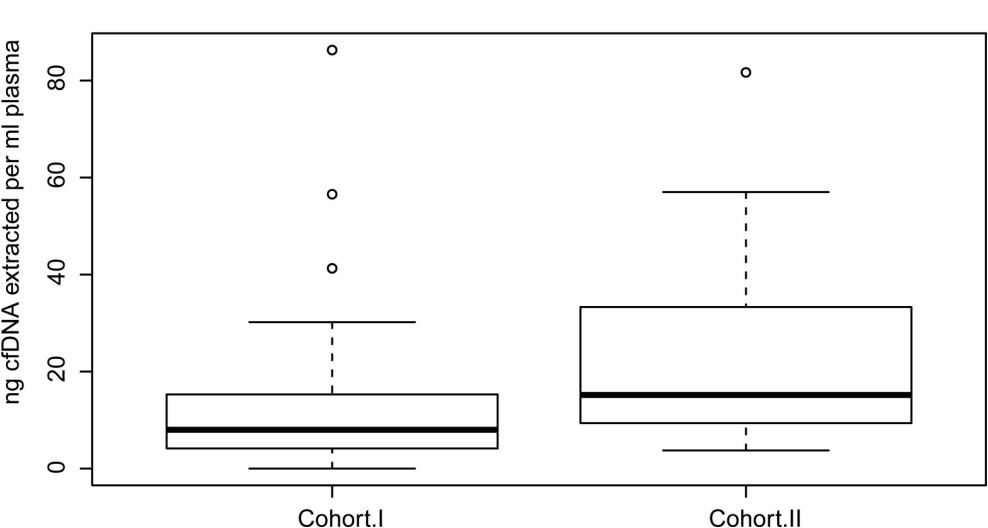

**Fig 6. Distribution of nanogram (ng) total cfDNA extracted per milliliter (ml) plasma from patients in cohort I (N = 31) and cohort II (N = 34).** The median amount of ng DNA per ml of plasma was 11.30 ng [2.33ng/ml;86.28ng/ml] for cohort I and 15.19 ng [3.75ng/ml;81.67ng/ml] for cohort II, respectively.

assays identified an EGFR p.L858R change in two cases considered WT by the Therascreen[®] assay (Tables 2 and S7A). Results obtained by the SensiScreen[®] EGFR Liquid assays were further validated and verified with 100% agreement to results obtained by the IOT[®] Oncomine cell-free nucleic acids assay (Thermo Fisher Scientific). This implies that the two samples found to be EGFR WT by the Therascreen[®] platform are EGFR p.L858R true positive.

For cohort II, as expected from the results previously obtained for sample 21 and 22 in the routine setting, together with the limited DNA amount available for test here, the latter resulted in the detection of only one of two EGFR variants by the SensiScreen[®] EGFR Liquid assay platform (Tables 3 and S7B). Hence, these two samples should be considered as agreement and not disagreement samples between platforms.

Further, with the ctEGFR Mutation Detection Kit (EntroGen), sample 11 was previously found positive for the c.2369C>T (p.T790M) variant and an exon 19 deletion using 18 ng of cfDNA per PCR reaction but only positive for the exon 19 deletion at 9 ng of cfDNA input in triplicate measurements (S1 Fig). Similarly, using 9 ng of cfDNA in single measurements, only the EGFR exon 19 deletion variant was detected by the SensiScreen[®] EGFR Liquid assay platform. Notably, absolute quantification suggested the presence of only 1–2 copies of the c.2369C>T (p. T790M) mutation variant in PCR reactions with DNA starting amounts of 18 ng performed during the initial routine analysis. Hence, it is likely that the 9 ng of sample 11 did not include any c.2369C>T (p.T790M) mutated DNA and therefore could not be detected by either the SensiScreen[®] EGFR Liquid assay (PentaBase ApS) or the ctEGFR Mutation Detection Kit (EntroGen).

For sample 31, the EGFR c.2369C>T (p.T790M) mutation was detected by the SensiScreen[®] EGFR Liquid assay platform (Tables 3 and S7B). Of major concern regarding routine analysis, an EGFR exon 19 deletion, but not the c.2369C>T (p.T790M) variant, was previously detected using the ctEGFR Mutation Detection Kit (EntroGen). Previously and prior to liquid biopsy analysis, EGFR solid tissue based mutational analysis had shown the presence of only an exon 19 deletion. Unfortunately, too small amounts of cfDNA was left for evaluation of the divergent results obtained for sample 11 and 31 (cohort II) using the IOT[®] Oncomine cell-free nucleic acids assay (Thermo Fisher Scientific) or another platform for test of liquid biopsies. However, further investigation of pleural fluid from the same patient collected fourteen weeks post-liquid biopsy analysis, showed the presence of both the c.2369C>T (p.T790M) and an exon 19 deletion.

## Discussion

In NSCLC, blood based liquid biopsy has found its way to the clinic as an alternative and supplement to tissue biopsy, primarily as a tool for identifying the mechanisms of resistance to

**Table 2. Overview of the EGFR phenotypes detected by the Therascreen[®] (QIAGEN) versus SensiScreen[®] EGFR Liquid assay (Pentabase) and IOT[®] Oncomine cell-free nucleic acids assay (Thermo Fisher Scientific) using cohort I and agreement or disagreement between platforms (N = 34).**

| TheraScreen[®] (QIAGEN) | SensiScreen[®] EGFR Liquid assay (PentaBase) IOT[®] Oncomine cell-free nucleic acids assay (Thermo Fisher Scientific) | | | | | | |
|---|---|---|---|---|---|---|---|
| | T790M & Ex19del. | T790M & L858R | WT | T790M | Ex19del. | L858R | L861Q |
| T790M & Ex19del. | 9 | - | - | - | - | - | - |
| T790M & L858R | - | 3 | - | - | - | - | - |
| WT | - | - | 13 | - | - | 2 | - |
| T790M | - | - | - | 1 | - | - | - |
| Ex19del. | - | - | - | - | 4 | - | - |
| L858R | - | - | - | - | - | 1 | - |
| L861Q | - | - | - | - | - | - | 1 |

Abbreviations: EGFR, Epidermal growth factor receptor; del, deletion; WT, wild type.

**Table 3. Overview of the EGFR phenotypes detected by the SensiScreen® EGFR Liquid assay (PentaBase ApS) versus the ctEGFR Mutation Detection Kit (Entro-Gen) using cohort II and agreement or disagreement between platforms (N = 34).**

| ctEGFR Mutation Detection Kit (EntroGen) | SensiScreen® EGFR Liquid (PentaBase) | | | | | |
|---|---|---|---|---|---|---|
| | T790M & Ex19del. | T790M & L858R | WT | T790M | Ex19del. | L858R |
| T790M & Ex19del. | 7 | - | - | - | [a&b]3 | - |
| T790M & L858R | - | 3 | - | - | - | - |
| WT | - | - | 9 | - | - | - |
| T790M | - | - | - | 4 | - | - |
| Ex19del. | - | - | - | [c]1 | 4 | - |
| L858R | - | - | - | - | - | 3 |

Abbreviations: EGFR, Epidermal growth factor receptor; ctEGFR, circulating tumor Epidermal growth factor receptor; del, deletion; WT, wild type.

targeted therapies and, secondly, to monitor the course of the disease. It is therefore important to have a reliable liquid biopsy test as the patient otherwise may be referred to an unpleasant tissue biopsy, if this is even feasible. For ctDNA analysis, there has been a diversity in terms of approaches utilized by laboratories making the ability to interpret and compare results across laboratories difficult [53]. Selection of approach is driven not only by needs, but also by available resources, and cost considerations. To date, the international agencies EMA and FDA do not specify which is the most sensitive methodology to analyse ctDNA, only generally recommending to implement a reliable test with high sensitivity (EMA website: http://www.ema.europa.eu). As both total cfDNA concentrations and the ratio of ctDNA/cfDNA extracted from blood samples are very low, it is of particularly importance to evaluate platform performance.

In the study we extracted cfDNA from two different tubes (streck vs citrate). It appears that citrate may give a statistically higher amount of DNA. A difference was observed between those two extraction methods (p = 0.021), with plasma citrate yielding higher amounts of DNA. This result might be due to the release of HMW DNA in the citrate cohort. It must therefore be considered to use plasma for future testing as amount of DNA may be a limitation (S1D Fig).

This study evaluated the performance of the newly developed and commercially available SensiScreen® EGFR Liquid kit for detection of the clinically relevant EGFR mutations in cfDNA from NSCLC patients—(c.2369C>T (p.T790M), Exon 19 deletions, c.2573T>G & c.2573_2574TG>GT (p.L858R)) -, accounting cumulatively more than 90% of the mutations that can hyper activate EGFR. The focus was particularly on performance in relation to low levels of ctDNA/cfDNA ratio input (<1%) (i.e. in the range of what can be extracted from human plasma and serum). At present, although international guidelines do not indicate a precise lowest value detectable, there is a consensus that every methodology applied to ctDNA must have a lowest value detectable <1%.

Initially, using samples with different concentrations of artificial templates (EGFR mutated DNA from cell lines) spiked into a background of WT DNA, we demonstrated—for the p.T790M, p.L858R, and exon 19 deletion assays—the possibility to detect as little as one copy (lowest value detectable = 0.06%) of mutant DNA in a WT background of approximately 1,600 copies using the SensiScreen® EGFR Liquid assays. Hence, the SensiScreen® EGFR Liquid assays are highly sensitive towards the most clinical important mutations found in EGFR even when using low amounts of DNA input (5 ng). The assessment of the LoB for each variant, through analyses of a pool of wt DNA from 24 healthy donors, permitted to confirm that, with SensiScreen® EGFR Liquid assays, we are able to detect one copy of the target DNA.

Furthermore, the three assays demonstrated high linearity ($R^2 \geq 0,97$) across a 7-log scale down to 0.1% of mutated copies. As expected, a higher degree of variation in recovery of fold changes across the seven log scale was observed at low mutant levels (0.5% and 0.1%), yet slopes of the best fitted line suggested high PCR efficiencies even at the inclusion of low mutant levels (0.5% and 0.1%), thus reinforcing the robustness of the assay.

As only a few mutant copies in a background of high amounts of total cfDNA (potentially coming from non-malignant cells elsewhere in the body) are likely to be found in liquid biopsy samples, assay repeatability tests detecting theoretically one copy of the mutant in WT DNA were performed. Importantly, all three assays showed repeatably positive detection in 75.5% - 80.4% of PCR reactions. We cannot fully determine if a potential difference from the expected (detection of theoretical one mutant copy) is due to pipetting errors or due to the assay or both. However, such observation is expected based on the laws of Poisson distribution and when our repeatability results are combined with the results of our sensitivity study, they do collectively indicate that the SensiScreen® EGFR Liquid Kit reliable detects a single copy of mutant. The repeatability experiment performed imitates the same situation as clinicians will face using clinical samples—i.e. DNA volumes added per PCR reaction may contain only one to few mutant copies in a background of WT DNA. Hence, nor will clinicians be able to conclude if a potential false negative result is due to Poison distribution, low ctDNA/cfDNA ratio, the assay or a combination. A way to circumvent this issue in the clinic is to perform more PCR replicates for a single liquid biopsy sample (as in our repeatability study) as the results demonstrates that two out of three samples comprising only one copy of mutated ctDNA is anticipated to be detected as positive with this setup. However, the drawback is the need to have more blood from the patients.

One of the major challenges is to have both high specificity and sensitivity also for the p. T790M change, that is very close to the polymorphism in codon 787. The p.T790M mutation has main clinical relevance, as it indicates resistance towards 1st and 2nd generation TKIs but sensitivity towards the 3rd generation TKI, osimertinib, which in addition also confers sensitivity against the activating EGFR mutations [13–16, 54, 55]. Competing RT-PCR assays (i.e. Cobas and Therascreen®) indicate a poorer lowest value detectable of the p.T790M variant as compared to the other EGFR mutations detected [56]. The utilization of the artificial DNA platform technology, INA®, and particularly the application of the BaseBlocker™ technology in the SensiScreen® EGFR Liquid Kit may collectively result in high specificity and sensibility also for the detection of the p.T790M variant, allowing for the detection of a single mutated copy of EGFR DNA. Multiple approaches are being pursued to bring input requirements down and to increase the signal-to-noise ratio by enhancement of the sensitivity and specificity—all because of the extremely low ctDNA levels in a background noise of cfDNA extracted from liquid biopsy samples [44, 56]. On the market, there are several methodologies characterized by different features: 1) extreme sensitivity when using high amounts of cfDNA but also high costs and the ability to investigate only one mutation at a time (ddPCR) vs 2) high sensitivity and the ability to investigate more mutations through comprehensive panels but expensive and time consuming (~2–3 days) with the requirement of special trained personnel (NGS) vs 3) high speed (~2 hours) and generally lower costs but with median sensitivity (RT-PCR). The SensiScreen® EGFR Liquid assay aims to combine the majority of positive features of the aforementioned approaches: high specificity and sensitivity, the possibility to investigate a diverse set of clinically relevant mutations at a time, high speed (less than two hours, from pipetting to the final result), low costs (similar to other RT-PCR based methods) and the possibility to be performed using different RT-PCR instruments. The high sensitivity and the lowest value detectable we observed may be explained by the INA® technology used in the SensiScreen® assay family (i.e. modification of the included oligonucleotides with pentabases).

Evaluation using clinical liquid biopsy samples was performed in two independent laboratories each with their own protocol from pre-analytical sample handling to RT-PCR instrumental setup and accredited for the normal diagnostic routine. In both laboratories, high agreement (94.1% - 100%) of the SensiScreen® EGFR Liquid assay to widely clinically used cfDNA platforms was observed. In our setup, the SensiScreen® EGFR Liquid assay platform (PentaBase) outperformed the Therascreen® platform (Qiagen) and showed 100% agreement at the sequence level to the IOT® Oncomine cell-free nucleic acids assay (Thermo Fisher Scientific). While the lowest value detectable of the Therascreen® is listed to 2% mutated alleles, both the SensiScreen® EGFR Liquid assay platform and the IOT® Oncomine cell-free nucleic acids assay have shown lowest value detectable of 0.1%—the difference of lowest values detectable among the methods may explain the observed discrepancies. Potentially, the SensiScreen® EGFR Liquid approach may be more robust and sensitive than the Therascreen® approach. Our results suggested that this is likely, at least for the p.L858R assay, as two samples found to be WT by the Therascreen® platform were detected as p.L858R mutation positive by the SensiScreen® EGFR Liquid and confirmed by the IOT® Oncomine cell-free nucleic acids assay. However, while the latter require 10 ng of cfDNA as input to obtain a lowest value detectable of 0.15%, the SensiScreen® EGFR Liquid assay platform requires two-fold less to obtain the same lowest value detectable. For some clinical samples with low cfDNA content per μl, it will not be possible to add 10 ng of cfDNA due to volume restrictions (i.e. <13 μl for the IOT® Oncomine cell-free nucleic acids assay). Hence, the lowest value detectable of 0.15% may not be obtained for all clinical samples.

In cohort II, the SensiScreen® EGFR Liquid assay platform (PentaBase) showed equal performance to the ctEGFR Mutation Detection Kit (EntroGen) in single replicate as compared to 2≤3 replicates for EntroGen (used in the routine laboratory at Herlev University Hospital). While the ctEGFR Mutation Detection Kit (EntroGen) is a multiplex assay constructed with allele specific primers targeting the c.2369C>T p.T790M variant, the c.2573T>G p.L858R variant, the 48 exon 19 aberrations and the beta-2 microglobulin (Internal control) in the same PCR reaction, the SensiScreen® EGFR Liquid assay is a four-assay based single and multiplex approach. This feature may be a limiting factor for samples with low cfDNA concentration as it may be difficult to fulfil the input requirements—both due to the volume restrictions per PCR reaction and due to the simplex construction between EGFR p.T790M, exon 19 deletions, p.L858R - as the extracted amount of cfDNA may be too low for some samples (the main factor is the requirement of cfDNA for four PCR reactions instead of one PCR reaction as for the ctEGFR Mutation Detection Kit (EntroGen).

## Conclusion

In our setup and clinical evaluation, the SensiScreen® EGFR Liquid assay platform (PentaBase) outperformed the Therascreen® platform (Qiagen) and showed equal performance to the IOT® Oncomine cell-free nucleic acids assay (Thermo Fisher Scientific) and the ctEGFR Mutation Detection Kit (EntroGen) using liquid biopsy plasma and serum samples. Additionally, we demonstrated that the SensiScreen® EGFR liquid assay robustly can detect a single copy of mutation (lowest value detectable = 0.06%), following the laws of Poisson distribution. The sensitivity and the lowest value detectable of the assay lie within the range of reported assay sensitivities (< 0.1 - < 1%) in the external quality assessment (EQA) scheme for ctDNA analysis [57]. Hence, the SensiScreen® EGFR Liquid assay may be a competitive platform and the platform of choice for cfDNA mutation profiling of p.T790M, p.L858R and exon 19 deletions in NSCLC.

## Supporting information

**S1 Table.** S1A Table. Clinical-pathological characteristics of cohort I used for analyses by the TheraScreen® (QIAGEN), the Ion Torrent® (Thermo Fisher Scientific) and the SensiScreen® EGFR Liquid assay (PentaBase) platforms. Cohort I was collected at the Institute of Pathology in Locarno, Switzerland. Abbreviations: AC, adenocarcinoma; F, female; M, male. S1B Table. Clinical-pathological characteristics of cohort II used for analyses by the ctEGFR Mutation Detection Kit (EntroGen) and the SensiScreen® EGFR Liquid assay (PentaBase). Cohort II was collected at the Department of Pathology, Herlev-Gentofte University Hospital, Denmark. Abbreviations: AC, adenocarcinoma; F, female; M, male.
(ZIP)

**S2 Table. Overview of the EGFR variant targets included in the SensiScreen® EGFR Liquid assay (Cat.no 5408 and 5409).** Abbreviations: CDS, coding sequence; del, deletion; EGFR, Epidermal growth factor receptor; ins, insertion.
(DOCX)

**S3 Table. PCR efficiency and the coefficient of determination for assays across seven- and five-log-scales.** Abbreviations: Ct, Cycle threshold; EGFR, Epidermal growth factor receptor.
(DOCX)

**S4 Table. Sensitivity average and standard deviation for assays based on different template concentrations.** Abbreviations: Ct, Cycle threshold; EGFR, Epidermal growth factor receptor, SD, Standard deviation.
(DOCX)

**S5 Table. Assessment of platform specificity for the SensiScreen® EGFR Liquid assays.** Ct values obtained when challenging the assays with either cell lines harbouring EGFR mutations or plasmid containing various mutated EGFR sequences diluted in WT cfDNA. Templates are analyzed in duplicates on the reference assay and in 4-double determination on the mutation specific assays. Abbreviations: Ct, Cycle threshold; EGFR, Epidermal growth factor receptor; NA, not available.
(DOCX)

**S6 Table. Statistical evaluation of repeatability calculated applying fractile limits according to theoretical number of zero samples and samples with more than 1 copy assuming a concentration of 1 copy/5 microL and Gaussian distribution.** Abbreviations: Ct, Cycle threshold; EGFR, Epidermal growth factor receptor.
(DOCX)

**S7 Table.** S7A Table. EGFR phenotypes obtained for cohort I as detected by the TheraScreen® (QIAGEN), the SensiScreen® EGFR Liquid assay (PentaBase) and the Ion Torrent® (Thermo Fisher Scientific) platforms. Abbreviations: del, deletion; EGFR, Epidermal growth factor receptor; IOT, Ion Torrent methodology; WT, wild type. aFor one of the three samples (Sample 11) with this combinatorial phenotype, using 9 ng of cfDNA as input for the ctEGFR Mutation Detection Kit (EntroGen) did only detect the EGFR exon 19 deletion variant, not the p.T790M variant. Triplicate measurements using 18 ng of cfDNA as input per PCR reaction and absolute quantification resulted in detection of the p.T790M variant in addition to the exon 19 deletion variant. From the standard curve, it was estimated that the copy number of the p.T790M variant was at the level 1–2 copies in 18 ng of cfDNA input. The SensiScreen® EGFR Liquid assay (PentaBase ApS) simplex did detect the exon 19 deletion using 9 ng of cfDNA per PCR reaction. bFor two of the three samples (Sample 21 & 22) with this

combinatorial phenotype, the DNA input amount available was too low for detection of the EGFR p.T790M variant by the SensiScreen® EGFR Liquid assay (PentaBase ApS) as previously evaluated from the low copy number of p.T790M detected at higher DNA input amounts by the ctEGFR Mutation Detection Kit (EntroGen). Hence, these two samples should be considered as agreement samples, both detecting the exon 19 deletion. cWhile the ctEGFR Mutation Detection Kit (EntroGen) detected an EGFR exon 19 deletion, the p.T790M variant but not the exon 19 deletion was detected by the SensiScreen® EGFR Liquid assay (PentaBase) (Sample 31). Duplicate measurements using the ctEGFR Mutation Detection Kit (EntroGen) showed two amplification curves for the p.T790M variant however below the threshold. S7B Table. Overview of the EGFR gene status for each plasma sample in cohort II as detected by the Entrogen ctEGFR Mutation Detection Kit (EntroGen) and the SensiScreen® EGFR Liquid assay (PentaBase ApS). Right column: EGFR gene status detected by previous analysis of tissue months to years prior to blood sampling for cfDNA EGFR analysis. aInitially for sample 11, triplicate measurements using 18 ng of cfDNA as input per PCR reaction resulted in detection of both the EGFR p.T790M and the exon 19 deletion variants. From standard curve analysis, it was estimated that the copy number of the p.T790M variant was 1–2 copies in 18 ng of cfDNA. Using 9 ng of cfDNA as input per PCR reaction for both the ctEGFR Mutation Detection Kit (EntroGen) (Triplicate measurements) and the SensiScreen® EGFR Liquid assay (PentaBase ApS) (single replicate) did only result in detection of the exon 19 deletion variant. Hence, it is likely that in 9 ng of cfDNA, no DNA copies of the c.2369C>T (p.T790M) variant was present and therefore could not be detected by either the SensiScreen® EGFR Liquid assay (PentaBase ApS) or the ctEGFR Mutation Detection Kit (EntroGen). bThe cfDNA amount available for test of the SensiScreen® EGFR Liquid assay (PentaBase ApS) was too low to detect the EGFR p.T790M variant as only few mutant copies were detected initially by the Entrogen ctEGFR Mutation Detection Kit (EntroGen). For sample 21, at 25 ng cfDNA input per PCR reaction used for the ctEGFR Mutation Detection Kit (EntroGen) the EGFR exon 19 deletion was detected, and the inclusion of copy number standards revealed that the p.T790M variant was estimated to be present in around 1–3 copies. Hence as evaluated from the copy number estimation using the ctEGFR Mutation Detection Kit (EntroGen) and with the lowest value detectable of the SensiScreen® EGFR Liquid assay (PentaBase ApS) in mind, it was expected that only the exon 19 deletion could be detected by the SensiScreen® EGFR Liquid assay at the 5 ng of cfDNA available for test. Using the SensiScreen® EGFR Liquid assay (PentaBase ApS), a low/late PCR amplification curve for the EGFR p.T790M variant was observed, however below the threshold. For sample 22, at 23 ng cfDNA as input per PCR reaction used for the ctEGFR Mutation Detection Kit (EntroGen) the EGFR exon 19 deletion was detected, and the inclusion of copy number standards revealed that the p.T790M variant was estimated to be present in around 1–2 copies. Only 10 ng of cfDNA was available for the test of the SensiScreen® EGFR Liquid assay (PentaBase ApS). Hence as evaluated from the copy number estimation using the ctEGFR Mutation Detection Kit (EntroGen) and with the lowest value detectable with the SensiScreen® EGFR Liquid assay (PentaBase ApS) in mind it was expected that only the exon 19 deletion may be detected by the SensiScreen® EGFR Liquid assay (PentaBase ApS). Using the SensiScreen® EGFR Liquid assay (PentaBase ApS), a low/late PCR amplification curve for the EGFR p.T790M variant was observed, however below the threshold. cFor sample 31, while the ctEGFR Mutation Detection Kit (EntroGen) detected an EGFR exon 19 deletion, the p.T790M variant but not the exon 19 deletion was detected by the SensiScreen® EGFR Liquid assay (PentaBase ApS). * For sample 13, TKI treatment may be based on inclusion in a clinical trial based on mutations in other genes than EGFR. Abbreviations: ctEGFR, circulating tumor Epidermal growth factor receptor; ctDNA, circulating tumor

DNA; del, deletion; EGFR, Epidermal growth factor receptor; WT, wild type.
(ZIP)

**S1 Fig. Detection of low copy numbers in sample 11 of cohort II using the ctEGFR Mutation Detection Kit (EntroGen).** A. Standard curve for the EGFR p.T790M target (EntroGen) constructed using the positive control provided with the ctEGFR Mutation Detection Kit (EntroGen). Four-fold template copy number differences was assessed using a six-log-scale ranging from theoretical 0,195 copy to 200 copies. B. Multicomponent plot of triplicate measurements for sample 11 (Cohort II) using 18 ng of cfDNA per PCR. As evaluated from time of detection (B) and compared to the standard curve (A), the absolute DNA copy numbers of the EGFR c.2369C>T (p.T790M) variant (Blue amplification curves) were estimated to be in the range 1–2 copies in 18 ng of cfDNA. C. Multicomponent plot of triplicate measurements for sample 11 (Cohort II) using 9 ng of cfDNA per PCR. Decreasing the amount of cfDNA to 9 ng confirmed that the copy numbers of the EGFR c.2369C>T (p.T790M) variant (Blue) was low for sample 11 (Cohort II) using 18 ng of cfDNA (B). B. and C.: Internal control primers amplify the beta-2 microglobulin (B2M) DNA in samples and are used both to determine the condition of reagents and whether the reaction contains sufficient amount of amplifiable DNA (Green amplification curves). Detection of EGFR exon 19 deletion variant (Purple amplification curves). Non-specific detection by the EGFR c.2573T>G p.L858R target (Red curves). Abbreviations: Ct, Cycle threshold; EGFR, Epidermal growth factor receptor. D: Results of the comparison between cfDNA extracted from blood collected from two different tubes (streck vs citrate). A difference was observed between those two extraction methods (p = 0.021), with plasma citrate yielding higher amounts of DNA.
(TIF)

## Author Contributions

**Conceptualization:** Steffen Grann Jensen, Samantha Epistolio, Rasmus Koefoed Petersen, Michael Børgesen, Ulf Bech Christensen, Estrid Høgdall, Milo Frattini.

**Data curation:** Steffen Grann Jensen, Samantha Epistolio, Cesilie Lind Madsen, Rasmus Koefoed Petersen, Michael Børgesen.

**Formal analysis:** Steffen Grann Jensen, Samantha Epistolio, Cesilie Lind Madsen, Rasmus Koefoed Petersen, Michael Børgesen.

**Investigation:** Steffen Grann Jensen, Samantha Epistolio, Cesilie Lind Madsen, Majbritt Hauge Kyneb, Alice Riva, Alessia Paganotti, Jessica Barizzi, Rasmus Koefoed Petersen, Michael Børgesen, Francesca Molinari, Renzo Boldorini, Jan Lorenzen, Erik Sørensen.

**Methodology:** Steffen Grann Jensen, Samantha Epistolio, Cesilie Lind Madsen, Rasmus Koefoed Petersen, Michael Børgesen, Erik Sørensen, Ulf Bech Christensen, Estrid Høgdall, Milo Frattini.

**Project administration:** Steffen Grann Jensen, Rasmus Koefoed Petersen.

**Resources:** Steffen Grann Jensen.

**Software:** Steffen Grann Jensen, Samantha Epistolio, Cesilie Lind Madsen.

**Supervision:** Steffen Grann Jensen.

**Validation:** Steffen Grann Jensen, Samantha Epistolio, Rasmus Koefoed Petersen, Michael Børgesen.

**Visualization:** Steffen Grann Jensen, Samantha Epistolio, Cesilie Lind Madsen, Rasmus Koefoed Petersen, Michael Børgesen.

**Writing – original draft:** Steffen Grann Jensen.

**Writing – review & editing:** Steffen Grann Jensen, Samantha Epistolio, Cesilie Lind Madsen, Majbritt Hauge Kyneb, Alice Riva, Alessia Paganotti, Jessica Barizzi, Rasmus Koefoed Petersen, Michael Børgesen, Francesca Molinari, Renzo Boldorini, Jan Lorenzen, Ulf Bech Christensen, Estrid Høgdall, Milo Frattini.

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
