## [Decision Letter · Decision Letter 0]

10 Aug 2020

PONE-D-20-13313

A new sensitive and fast assay for the detection of EGFR mutations in liquid biopsies

PLOS ONE

Dear Dr. Jensen,

Thank you for submitting your manuscript to PLOS ONE. After careful consideration, we feel that it has merit but does not fully meet PLOS ONE’s publication criteria as it currently stands. Therefore, we invite you to submit a revised version of the manuscript that addresses the points raised during the review process.

We look forward to receiving your revised manuscript.

Kind regards,

Amit Dutt, Ph.D

Academic Editor

PLOS ONE

Additional Editor Comments:

Please refer to the attached file for the second reviewer's comments.

Journal Requirements:

"Competing Interests:

UBC, RKP, MB, and CLM are employees of PentaBase ApS.

The SensiScreen Liquid EGFR assay is now part of a marketed product portfolio of PentaBase ApS." 

We note that one or more of the authors are employed by a commercial company: PentaBase ApS.

2.1. Please provide an amended Funding Statement declaring this commercial affiliation, as well as a statement regarding the Role of Funders in your study. If the funding organization did not play a role in the study design, data collection and analysis, decision to publish, or preparation of the manuscript and only provided financial support in the form of authors' salaries and/or research materials, please review your statements relating to the author contributions, and ensure you have specifically and accurately indicated the role(s) that these authors had in your study. You can update author roles in the Author Contributions section of the online submission form.

2.2. Please also provide an updated Competing Interests Statement declaring this commercial affiliation along with any other relevant declarations relating to employment, consultancy, patents, products in development, or marketed products, etc. 

Reviewers' comments:

Reviewer's Responses to Questions

**Comments to the Author**

1. Is the manuscript technically sound, and do the data support the conclusions?

Reviewer #1: Yes

Reviewer #2: Partly

2. Has the statistical analysis been performed appropriately and rigorously? 

Reviewer #1: Yes

Reviewer #2: No

3. Have the authors made all data underlying the findings in their manuscript fully available?

Reviewer #1: Yes

Reviewer #2: Yes

4. Is the manuscript presented in an intelligible fashion and written in standard English?

Reviewer #1: Yes

Reviewer #2: Yes

5. Review Comments to the Author

Reviewer #1: 1. This study evaluated the performance of the newly developed and commercially available SensiScreen® EGFR Liquid kit for detection of the most diffused and clinically relevant EGFR mutations in cfDNA from NSCLC patients - (c.2369C>T (p.T790M), Exon 19 deletions, c.2573T>G & c.2573_2574TG>GT (p.L858R)) -, accounting cumulatively more than 90% of the mutations that can hyper activate EGFR. The results found that the SensiScreen® EGFR Liquid assay is able to identify down to 1 copy of mutant alleles and with its high sensitivity, linearity and accuracy it may be a competitive platform of choice.

2. Experiments, statistics, and other analyses are performed to a high technical standard and are described in sufficient detail.

3. Conclusions are presented in an appropriate fashion and are supported by the data.

Reviewer #2: The authors should be commended on performing this study which defines the performance of a commercially available test which will be used to aid physicians in understanding the course of care for their patient. The strength of the paper is in the orthogonal testing done to support the clinical observations of the test in addition to the repeatability and challenge of the low end sensitivity of the test. The weakness in the paper is the lack of items in the analytical validation work, specifically in the lack of testing variant negative patient samples to establish the LoB and the clinical specificity of the test. This reviewer recommends that the authors follow the conventions of the analytical assessments of a clinical diagnostic with their use of terminology and how these are measured and displayed, such as LoD and linearity.

I would like to see the authors discuss a potential weakness of the assay in the wide allowance of amplifiable DNA used in the test and how this wide window may result in inconsistent clinical assessments of the EGFR mutation.

Please add some statistical test to groups you display in graphs and also add ranges to your data and 95% CI as necessary.

Overall it is recommended that this paper be published in PLoS One as it is important that assessments of tools being used by clinicians be properly described and compared.

Please see attached review.

6. PLOS authors have the option to publish the peer review history of their article (what does this mean?). If published, this will include your full peer review and any attached files.

Reviewer #1: No

Reviewer #2: No

---

## [Author Response · Author response to Decision Letter 0]

20 Apr 2021

Please find the point by point response to editor and to reviewers in the file attached and named "Response to reviewers".

Thank you for your revisions and comments.

Kind regards.

Dr. Samantha Epistolio, PhD

---

## [Decision Letter · Decision Letter 1]

11 Jun 2021

A new sensitive and fast assay for the detection of EGFR mutations in liquid biopsies

PONE-D-20-13313R1

Dear Dr. Epistolio,

We’re pleased to inform you that your manuscript has been judged scientifically suitable for publication and will be formally accepted for publication once it meets all outstanding technical requirements.

Kind regards,

Jeffrey Chalmers, Ph.D.

Academic Editor

PLOS ONE

Additional Editor Comments (optional):

Reviewers' comments:

Reviewer's Responses to Questions

**Comments to the Author**

1. If the authors have adequately addressed your comments raised in a previous round of review and you feel that this manuscript is now acceptable for publication, you may indicate that here to bypass the “Comments to the Author” section, enter your conflict of interest statement in the “Confidential to Editor” section, and submit your "Accept" recommendation.

Reviewer #1: All comments have been addressed

Reviewer #2: All comments have been addressed

2. Is the manuscript technically sound, and do the data support the conclusions?

Reviewer #1: Yes

Reviewer #2: Yes

3. Has the statistical analysis been performed appropriately and rigorously? 

Reviewer #1: Yes

Reviewer #2: Yes

4. Have the authors made all data underlying the findings in their manuscript fully available?

Reviewer #1: Yes

Reviewer #2: (No Response)

5. Is the manuscript presented in an intelligible fashion and written in standard English?

Reviewer #1: Yes

Reviewer #2: Yes

6. Review Comments to the Author

Reviewer #1: All raised points have been adequately addressed and English language and style are fine, but a minor spell check is required.

Reviewer #2: Excellent revision of the previous manuscript. All concerns were addressed and limitations were discussed in a thorough and thoughtful manner. This is a good example of a well researched evaluation of a molecular test which will be utilized to aid physicians in their treatment decisions for patient who may benefit from EGFR TKIs.

7. PLOS authors have the option to publish the peer review history of their article (what does this mean?). If published, this will include your full peer review and any attached files.

Reviewer #1: **Yes: **Rui-hua Xu

Reviewer #2: No

---

## [Editor Report · Acceptance letter]

15 Jun 2021

PONE-D-20-13313R1 

A new sensitive and fast assay for the detection of EGFR mutations in liquid biopsies 

Dear Dr. Epistolio:

I'm pleased to inform you that your manuscript has been deemed suitable for publication in PLOS ONE. Congratulations! Your manuscript is now with our production department. 

Kind regards, 

on behalf of

Dr. Jeffrey Chalmers 

Academic Editor

PLOS ONE